# Reinforcement Learning from Reformulations in Conversational Question Answering over Knowledge Graphs

Magdalena Kaiser
Max Planck Institute for Informatics
mkaiser@mpi-inf.mpg.de

Rishiraj Saha Roy
Max Planck Institute for Informatics
rishiraj@mpi-inf.mpg.de

Gerhard Weikum
Max Planck Institute for Informatics
weikum@mpi-inf.mpg.de

## ABSTRACT

The rise of personal assistants has made conversational question answering (ConvQA) a very popular mechanism for user-system interaction. State-of-the-art methods for ConvQA over knowledge graphs (KGs) can only learn from crisp question-answer pairs found in popular benchmarks. In reality, however, such training data is hard to come by: users would rarely mark answers explicitly as correct or wrong. In this work, we take a step towards a more natural learning paradigm – from noisy and implicit feedback via question reformulations. A reformulation is likely to be triggered by an incorrect system response, whereas a new follow-up question could be a positive signal on the previous turn's answer. We present a reinforcement learning model, termed CONQUER, that can learn from a conversational stream of questions and reformulations. CONQUER models the answering process as multiple agents walking in parallel on the KG, where the walks are determined by actions sampled using a policy network. This policy network takes the question along with the conversational context as inputs and is trained via noisy rewards obtained from the reformulation likelihood. To evaluate CONQUER, we create and release CONVREF, a benchmark with about $11k$ natural conversations containing around $205k$ reformulations. Experiments show that CONQUER successfully learns from noisy rewards, significantly improving over a state-of-the-art baseline.

## CCS CONCEPTS

• **Information systems → Question answering**;

## KEYWORDS

Question Answering, Knowledge Graphs, Conversations, Feedback

**ACM Reference Format:**
Magdalena Kaiser, Rishiraj Saha Roy, and Gerhard Weikum. 2021. Reinforcement Learning from Reformulations in Conversational Question Answering over Knowledge Graphs. In *Proceedings of the 44th International ACM SIGIR Conference on Research and Development in Information Retrieval (SIGIR '21), July 11–15, 2021, Virtual Event, Canada.* ACM, New York, NY, USA, 11 pages. https://doi.org/10.1145/3404835.3462859

## 1 INTRODUCTION

**Background and motivation.** Conversational question answering (ConvQA) has become a convenient and natural mechanism of

satisfying information needs that are too complex or exploratory to be formulated in a single shot [11, 23, 30, 47, 50, 52]. ConvQA operates in a multi-turn, sequential mode of information access: utterances in each turn are ad hoc and often incomplete, with implicit context that needs to be inferred from prior turns. When the information needs are fact-centric (e.g., about cast of movies, clubs of soccer players, etc.), a suitable data source to retrieve answers from are large knowledge graphs (KG) such as Wikidata [65]. Fig. 1 shows a small excerpt of the KG using a simplified graph representation, with red nodes for entities and blue nodes for relations. We address ConvQA over KGs, where system responses are entities.

**Example.** An ideal conversation with five *turns* could be as follows ($q_i$ and $ans_i$ are questions and answers at turn $i$, respectively):

> $q_1$: *When was Avengers: Endgame released in Germany?*
> $ans_1$: *24 April 2019*
> $q_2$ : *What was the next from Marvel?*
> $ans_2$: *Spider-Man: Far from Home*
> $q_3$ : *Released on?*
> $ans_3$: *04 July 2019*
> $q_4$: *So who was Spidey?*
> $ans_4$: *Tom Holland*
> $q_5$ : *And his girlfriend was played by?*
> $ans_5$: *Zendaya Coleman*

Utterances can be colloquial ($q_4$) and incomplete ($q_2, q_3$), and inferring the proper context is a challenge ($q_5$). Users can provide *feedback* in the form of *question reformulations* [44]: when an answer is incorrect, users may rephrase the question, hoping for better results. While users never know the correct answer upfront, they may often guess non-relevance when the answer does not match the expected type (director instead of movie) or from additional background knowledge. So, in reality, turn 2 in the conversation above could become expanded into:

> $q_{21}$ : *What was the next from Marvel?* **(New intent)**
> $ans_{21}$: *Stan Lee* **(Wrong answer)**
> $q_{22}$ : *What came next in the series?* **(Reformulation)**
> $ans_{22}$: *Marvel Cinematic Universe* **(Wrong answer)**
> $q_{23}$: *The following movie in the Marvel series?* **(Reformulation)**
> $ans_{23}$: *Spider-Man: Far from Home* **(Correct answer)**
> $q_{31}$: *Released on?* **(New intent)**

**Limitations of state-of-the-art.** Research on ConvQA over KGs is still in its infancy [12, 23, 52, 55] – in particular, there is virtually no work on considering user signals when intermediate utterances lead to unsatisfactory responses as indicated in the above conversation with reformulations. A few works on QA over KGs has exploited user interactions for online learning [2, 72], but this is limited to confirming the correctness of answers which can then augment the training data of question-answer pairs. Reformulations

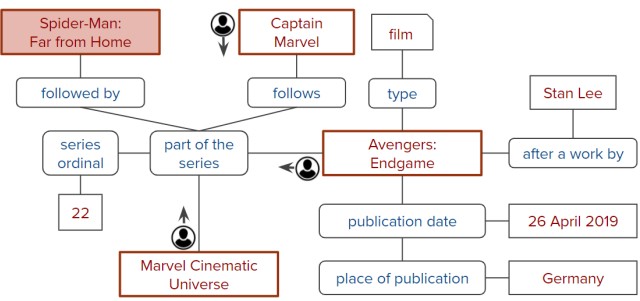

**Figure 1: KG excerpt from Wikidata required for answering $q_1$ and $q_2$. Agents for $q_{22}$ (*What came next in the series?*) are shown with possible walk directions. The colored box (`Spider-man: Far from Home`) is the correct answer.**

as an implicit feedback signal have been leveraged for web search queries [29, 51], but that setting is very different from QA over KGs. IR methods rely on observing clicks (and their absence) on ranked lists of documents. This does not carry over to typical QA tasks – especially over voice interfaces with single-entity responses at each turn and no explicitly positive click-like signal.

**Approach.** We present Conquer (Conversational Question answering with Reformulations), a new method for learning from implicit user feedback in ConvQA over KGs. Conquer is based on reinforcement learning (RL) and is designed to continuously learn from question reformulations as a cue that the previous system response was unsatisfying. RL methods have been pursued for KG reasoning and multi-hop QA over KGs [15, 35, 46]. However, *conversational* QA is very different from these prior setups.

Given the current (say $q_{22}$) and the previous utterances ($q_1, q_{21}$), Conquer creates and maintains a set of *context entities* from the KG that are the most relevant to the conversation so far. It then positions *RL agents* at each of these context entities, that simultaneously walk over the KG to other entities in their respective neighborhoods. End points of these walks are candidate answers for this turn and are aggregated for producing the final response. Walking directions (see arrows in Fig. 1 for illustration) are decided by sampling actions from a policy network that takes as input i) encodings of utterances, and ii) KG facts involving the context entities. The policy network is trained via noisy rewards obtained from reformulation likelihoods estimated by a fine-tuned BERT predictor. Experiments on our ConvRef benchmark, that we created from conversations between a system and real users, demonstrate the viability of our proposed learning method Conquer and its superiority over a state-of-the-art baseline. The benchmark, a demo, and the link to our code on GitHub are publicly accessible at https://conquer.mpi-inf.mpg.de.

**Contributions.** Salient contributions of this work are:

- A question answering method that can learn from a conversational stream in the *absence of gold answers*;
- A *reinforcement learning* model for QA with rewards based on implicit feedback in the form of question *reformulations;*
- A *reformulation detector* based on BERT that can classify a follow-up utterance as a reformulation or new intent;
- A new *benchmark collection with reformulations* for ConvQA over KGs, comprising about 11$k$ conversations with more than 200$k$ turns in total, out of which 205$k$ are reformulations.

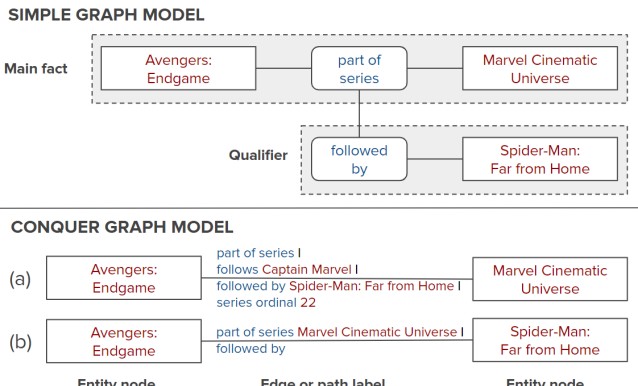

**Figure 2: Conquer KG representation for using qualifiers.**

## 2 MODEL AND ARCHITECTURE

### 2.1 Conquer KG representation

**Basic model.** A knowledge graph (KG) $K$ is typically stored as an RDF database organized into $\langle S, P, O \rangle$ (subject, predicate, object) triples (facts), where $S$ is an entity (e.g., `Avengers: Endgame`, `Stan Lee`), $P$ is a predicate (e.g., `part of series`, `publication date`), and $O$ is an entity, a type (e.g., `film`, `country`) or a literal (e.g., `26 April 2019`, `22`). In Conquer, we wish to leverage the entire KG for answering. For that we need to go beyond triples and consider *n-ary facts*, as discussed below.

**Qualifier model.** Large KGs like Wikidata also contain *n*-ary facts that involve more than two entities. Examples are: cast information involving a movie, a character role and a cast member or movie trilogy information requiring the movies, their ordinal numbers and the name of the series. *n*-ary facts are typically represented as a *main fact* enhanced with *qualifiers*, that are (possibly multiple) auxiliary ⟨predicate, object⟩ pairs adding contextual information to the main fact. For example, in Fig. 1, the path in the graph connecting `Avengers: Endgame` to `part of the series` and over to `Marvel Cinematic Universe` represent a main fact, which is contextualized by the path connecting this main fact to the qualifier predicate node `followed by` and on to the qualifier object node `Spider-man: Far from Home`. The path with `series ordinal` and `22` is another qualifier for the same main fact. The KG schema determines which part of the fact is considered main and which part a qualifier. This is often interchangeable, and in practice each provides context for the other. A large part of QA research disregards qualifiers, but they contain valuable information [21, 26, 33, 40, 42] and constitute a substantial fraction of Wikidata and other KGs. Without considering qualifiers in Wikidata, we would not be able to answer *any of the questions from $q_1$ through $q_5$.*

In the graph representation depicted in Fig. 1, qualifier predicates are directly connected to their main-fact predicates. However, in this representation, an agent walking from entity to entity (e.g., `Avengers` to `Spider-man`) misses the context of `Marvel` and would be useless for answering $q_2$. In this case, the main-fact triple (<`Avengers: Endgame`, `part of series`, `Marvel Cinematic Universe`>) provides necessary context for making sense of the qualifier (<`followed by`, `Spider-Man: Far from Home`>). To take *n*-ary facts into account during walks by agents, Conquer creates a modified KG representation where

| Notation | Concept |
|---|---|
| $K$ | Knowledge graph |
| $q, ans$ | Question and answer |
| $C$ | Conversation |
| $I$ | Intent |
| $q_{j1}$ | First question in intent $j$ |
| $\langle q_{jk}|k > 1\rangle$ | Sequence of reformulations in intent $j$ |
| $t$ | Turn |
| $q_t^{cxt} \in Q_t^{cxt}$ | Context questions at turn $t$ |
| $e_t^{cxt} \in E_t^{cxt}$ | Context entities at turn $t$ |
| $h_{(\cdot)}$ | Hyperparameters for context entity selection |
| $\boldsymbol{q}_t, \boldsymbol{q}_t^{cxt}, \boldsymbol{e}_t^{cxt}$ | Embedding vectors of $q_t, q_t^{cxt}, e_t^{cxt}$ |
| $s \in \mathcal{S}$ | RL states |
| $a \in A_s$ | Actions at state $s$ |
| $\boldsymbol{a}, \boldsymbol{A}_s$ | Embedding vector of $a$, and matrix of all actions at $s$ |
| $p \in \mathcal{P}$ | Path labels in $K$ |
| $R$ | Reward |
| $\boldsymbol{\theta}$ | Parameters of policy network |
| $\pi_{\boldsymbol{\theta}}$ | Policy parameterized by $\boldsymbol{\theta}$ |
| $J(\boldsymbol{\theta})$ | Expected reward with $\boldsymbol{\theta}$ |
| $\boldsymbol{W}_1, \boldsymbol{W}_2$ | Weight matrices in policy network |
| $\alpha$ | Step size in REINFORCE update |
| $H_\pi(\cdot, s)$ | Entropy regularization term in REINFORCE update |
| $\beta$ | Weight for entropy regularization term |
| $e^{ans}$ | Candidate answer entity |

Table 1: Notation for salient concepts in CONQUER.

entities are nodes and edges between entities are labeled either by connecting predicates (when a fact has no qualifiers, like <Avengers: Endgame, after a work by, Stan Lee>) or by *augmented labels* in cases of facts with qualifiers. The latter scenario is visualized in Fig. 2. The edge between the main-fact subject Avengers: Endgame and the main-fact object Marvel Cinematic Universe is augmented by its qualifier information in Fig. 2 (a). Information from the main fact is also used to augment the connections between the main-fact subject (or object) and qualifier objects, as in Fig. 2 (b). Connections between qualifier objects are analogously augmented by the main fact. These edge labels or *paths* subsequently become *actions* to be chosen by RL agents. The CONQUER graph model is *bidirectional*.

## 2.2 ConvQA concepts

We now define key concepts for ConvQA below. A notation overview is in Table 1 (some concepts are introduced only in later sections).
**Question**. A question $q$ (aka utterance) is composed of a sequence of words that is issued by the user to instantiate a specific information need in the conversation. We make no assumptions on the grammatical correctness of $q$. Questions may express new information needs or reformulate existing ones.
**Answer**. An answer $ans$ is a single or a (typically small) set of entities (or literals) from $K$, that the system returns to the user in response to her question $q$.
**Conversation**. A conversation $C$ is a sequence of questions $\langle q \rangle$ and corresponding answers $\langle ans \rangle$. $C$ can be perceived as being organized into a sequence of user intents.
**Intent**. Each distinct information need in a conversation $C$ is referred to as an intent $I$. The ideal conversation in Sec. 1 has five intents $\langle I_1, \ldots, I_5 \rangle$. Intents are latent and expressed by questions $q$.

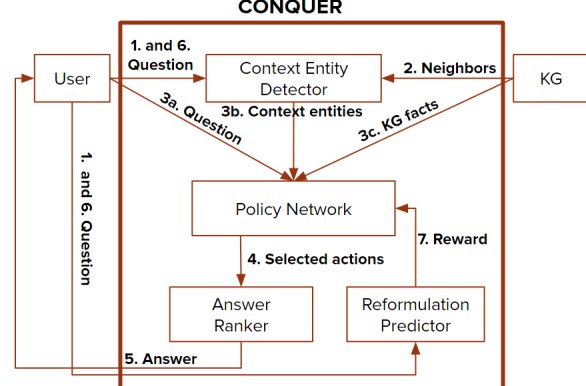

Figure 3: Overview of CONQUER, with numbered steps tracing the workflow. Looping through steps 1-7 creates a continuous learning model. Step 6 denotes a follow-up question.

**Reformulation**. For a specific intent $I_j$, a user issues reformulation questions $\langle q_{jk}|k > 1\rangle$ ($q_{j2}, q_{j3}, \ldots$) when the system response $ans$ to the first question $q_j$ (equivalently $q_{j1}$) was wrong. All intents, including the first one, can be potentially reformulated.
**Turn**. Each question in $C$, including its reformulations and corresponding answers, constitutes a turn $t_i$. For instance, for conversation $C = \langle q_{11}, ans_{11}, q_{12}, ans_{12}, q_{21}, ans_{21}, q_{22}, ans_{22}, q_{31}, ans_{31} \rangle$, we have three intents ($I_1, I_2, I_3$), five questions $q_{(\cdot)}$, five answers $ans_{(\cdot)}$, two reformulations ($q_{12}, q_{22}$), and *five* turns ($t_1, \ldots, t_5$). Thus, $C$ may also be written as $\langle q_{t_1}, ans_{t_1}, q_{t_2}, ans_{t_2}, q_{t_3}, ans_{t_3}, \ldots, q_{t_5}, ans_{t_5} \rangle$. To simplify notation, when we refer to a question at turn $t_i$, we will only use $q_i$ instead of $q_{t_i}$ (analogously for context).
**Context questions**. At any given turn $t$, context questions $Q_t^{cxt}$ are the ones most relevant to the conversation so far. This set may be comprised of a few of the immediately preceding questions $q_{t-1}, q_{t-2}, \ldots$, or include the first question ($q_1$) in $C$ as well.
**Context entities**. At any given turn $t$, context entities $E_t^{cxt}$ are the ones most relevant to the conversation so far. These are identified using various cues from the question and the KG and form the *start points* of the walks by the RL agents.

## 2.3 System overview

The workflow of CONQUER is illustrated in Fig. 3. First, context entities up to the current turn of the conversation are identified. Next, paths from our KG model involving these entities are extracted, and *RL agents* walk along these paths to candidate answers. The paths to walk on (actions by the agent) are decided according to predictions from a *policy network*, which takes as input the conversational context and the KG paths. Aggregating *end points* of walks by the different agents leads to the final answer. Upon observing this answer, the user issues a *follow-up question*. A *reformulation predictor* takes this <original question, follow-up question> sequence as input and outputs a reformulation likelihood. Parameters of the policy network are then updated in an online manner using rewards that are based on this likelihood. Context entities are reset at the end of the conversation, but the policy parameters continue to be updated as more and more conversations take place between the user and the system. Sec. 3 through 5 describe CONQUER in detail.

## 3 DETECTING CONTEXT ENTITIES

Throughout a conversation $C$, we maintain a set of context entities $E^{cxt}$ that reflect the user's topical focus and intent. For full-fledged questions this would call for Named Entity Disambiguation (NED), linking entity mentions onto KG nodes [56]. There are many methods and tools for this purpose, including a few that are geared for very short inputs like questions [34, 43, 54]. However, none of these can handle contextually incomplete and colloquial utterances that are typical for follow-up questions in a conversation, for example: *What came next from Marvel?* or *Who played his girlfriend?*. The set $E^{cxt}$ is comprised of the relevant KG nodes for the conversation so far and is created and maintained as follows.

The set $E^{cxt}$ emanates from the question keywords and is initialized by running an NED tool on the *first question* $q_1$, which is almost always well-formed and complete. Further turns in the conversation incrementally augment the set $E^{cxt}$. Correct answers for questions could qualify for $E^{cxt}$ and would be strong cues for keeping context. However, they are not considered by CONQUER: the online setting that we tackle does not have any knowledge of ground-truth answers and would thus have to indiscriminately pick up both correct and incorrect answers. Therefore, CONQUER considers only entities derived from user utterances.

Let $E_{t-1}^{cxt}$ denote the set of context entities up to turn $t-1$. Nodes in the neighborhood $nbd(\cdot)$ of $E_{t-1}^{cxt}$ form the *candidate context* for turn $t$ and are subsequently scored for entry into $E_t^{cxt}$. In our experiments, we restrict this to 1-hop neighbors, which is usually sufficient to capture all relevant cues. For scoring candidate entities $n$ for question $q_t$ at turn $t$, CONQUER computes four measures for each $n \in nbd(e|e \in E_{t-1}^{cxt})$:

- **Neighbor overlap:** This is the number of nodes in $E_{t-1}^{cxt}$ from where $n$ is reachable in one hop, the higher the better. Since this indicates high connectivity to $E_{t-1}^{cxt}$, such nodes are potentially good candidates. The number is normalized by the cardinality $|E_{t-1}^{cxt}|$, to produce $overlap(n) \in [0, 1]$.

- **Lexical match:** This is the Jaccard overlap between the set of words in the node label of $n$ and all words in $q_t$ (with stopwords excluded): $match(n) \in [0, 1]$.

- **NED score:** Although full-fledged NED would not work well for incomplete and colloquial questions, NED methods can still give useful signals. We run an off-the-shelf tool, which we provide with richer context by concatenating the current with the previous questions as input. We consider its normalized confidence score, $ned(n) \in [0, 1]$, but only if the returned entity is in the candidate set $nbd(e|e \in E_{t-1}^{cxt})$; otherwise $ned(n)$ is zero. This can be thought of as NED restricted to the neighborhood of the current context $E^{cxt}$ as an entity repository.

- **KG prior:** Salient nodes in the KG, as measured by the number of facts they are present in as subject, are indicative of their importance in downstream tasks like QA [12]. A prior on this KG frequency often helps discriminate obscure nodes from prominent ones. We clip raw frequencies at a factor $f_{max}$, and normalize them by $f_{max}$ to yield $prior(n) \in [0, 1]$.

These four scores are linearly combined with hyperparameters $h_1, \ldots, h_4$, such that $\sum_{i=1}^4 h_i = 1$, to compute the *context score*:

$$cxt(n) = h_1 \cdot overlap(n) + h_2 \cdot match(n) + h_3 \cdot ned(n) + h_4 \cdot prior(n) \quad (1)$$

If score $cxt(n)$ is above a specified threshold $h_{cxt}$, then $n$ is inserted into the set of context entities $E_t^{cxt}$. Hyperparameters $h_1, \ldots, h_4$ and $h_{cxt}$ are tuned on a development set. Entities in $E_t^{cxt}$ are passed on as start points for RL agents to walk from (Sec. 4.2).

## 4 LEARNING FROM REFORMULATIONS

### 4.1 RL Model

The goal of an RL agent here is to learn to answer conversational questions correctly. The user (issuing the questions) and the KG jointly represent the environment. The agent walks over the knowledge graph $K$, where entities are represented as nodes and predicates as path labels (Sec. 2.1). An agent can only start and end its walk at entity nodes (Fig. 2), after traversing a path label. This traversal can be viewed as a Markov Decision Process (MDP), where individual parts $(S, A, \delta, R)$ are defined as follows (adapted from [15]):

- **States:** A state $s \in S$ is represented by $s = (q_t^{cxt}, q_t, e_t^{cxt})$, where $q_t$ represents the question at turn $t$ (new intent or reformulation), $q_t^{cxt} \in Q_t^{cxt}$ captures a subset of the previous utterances as the (optional) context questions and $e_t^{cxt} \in E_t^{cxt}$ is one of the context entities for turn $t$ that serves as the starting point for an agent's walk.

- **Actions:** The set of actions $A_s$ that can be taken in state $s$ consists of all outgoing paths of the entity node $e_t^{cxt}$ in $K$, so that $A_s = \{p | \langle e_t^{cxt}, p, e^{ans} \rangle \in K\}$. End points of these paths are candidate answers $e^{ans}$.

- **Transitions:** The transition function $\delta$ updates a state to the agent's destination entity node $e^{ans}$ along with the follow-up question and (optionally) its context questions; $\delta : S \times A \rightarrow S$ is defined by $\delta(s, a) = s_{next} = (q_{t+1}^{cxt}, q_{t+1}, e^{ans})$.

- **Rewards:** The reward depends on the next user utterance $q_{t+1}$. If it expresses a new intent, then reward $R(s, a, s_{next}) = 1$. If $q_{t+1}$ has the same intent, then this is a reformulation, making $R(s, a, s_{next}) = -1$.

While we know deterministic transitions inside the KG through nodes and path labels, users' questions are not known upfront. So we use a *model-free algorithm* that does not require an explicit model of the environment [59]. Specifically, we use Monte Carlo methods that rely on explicit trial-and-error *experience*: we learn from sampled state-action-reward sequences from actual or simulated interaction with the environment. Since questions can be arbitrarily formulated on any of several topics, our state space is unbounded in size. Thus, it is not feasible to learn transition probabilities between states. Instead, we use a *parameterized policy* that learns to capture similarities between the question (along with its conversational context) and the KG facts. The parameterized policy is manifested in the weights of a neural network, called the *policy network*. When a new question arrives, this policy can be applied by an agent to reach an answer entity: this is equivalent to the agent following a path predicted by the policy network.

### 4.2 RL Training

Using a policy network has been shown to be more appropriate for KGs due to the large action space [67]. The alternative of using value functions (e.g. Deep Q-Networks [38]) may have poor convergence in such scenarios. To train our network, we apply the

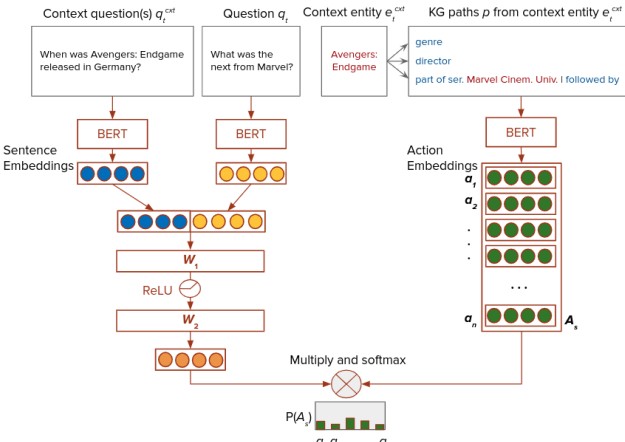

**Figure 4: Architecture of the policy network in CONQUER.**

policy-gradient algorithm REINFORCE with baseline [66]. As baseline, we use the average reward over several training samples for variance reduction. The parameterized policy $\pi_\theta$ takes information about a state as input and outputs a probability distribution over the available actions in this state. Formally: $\pi_\theta(s) \mapsto P(A_s)$.

Fig. 4 depicts our policy network which contains a two-layer feed-forward network with non-linear ReLU activation. The policy parameters $\theta$ consist of the weight matrices $W_1$, $W_2$ of the feed-forward layers. Inputs to the network consist of embeddings of the current question $q_t \in \mathbb{R}^d$ (*What was the next from Marvel?*), optionally prepended with some context question embeddings $q_t^{cxt} \in \mathbb{R}^d$ (like *When was Avengers: Endgame released in Germany?*). We apply a pre-trained BERT model to obtain these embeddings by averaging over all hidden layers and over all input tokens. Context entities $E_t^{cxt}$ (e.g., Avengers: Endgame) are the starting points for an agent's walk and are identified as explained in Sec. 3. We then retrieve all outgoing paths for these entities from the KG. An action vector $a \in A_s$ consists of the embedding of the respective path $p$ starting in $e_t^{cxt}$, $a \in \mathbb{R}^d$. These actions are also encoded using BERT. The final embedding matrix $A_s \in \mathbb{R}^{|A_s| \times d}$ consists of the stacked action embeddings. The output of the policy network is the probability distribution $P(A_s)$, that is defined as follows:

$$P(A_s) = \sigma(A_s \times (W_2 \times ReLU(W_1 \times [q_t^{cxt}; q_t]))) \quad (2)$$

where $\sigma(\cdot)$ is the softmax operator. Then, the final action which the agent will take in this step is sampled from this distribution:

$$a = A_s^i, \; i \sim Categorical(P(A_s)) \quad (3)$$

To update the network's parameters $\theta$, the expected reward $J(\theta)$ is maximized over each state and the corresponding set of actions:

$$J(\theta) = \mathbb{E}_{s \in S} \mathbb{E}_{a \sim \pi_\theta}[R(s, a, s_{next})] \quad (4)$$

For each question in our training set, we do multiple *rollouts*, meaning that the agent samples multiple actions for a given state to estimate the stochastic gradient (the inner expectation in the formula above). Updates to our policy parameters $\theta$ are performed in batches. Each *experience* of the form $(s, a, s_{next}, R)$ that the agent has encountered is stored. A batch of experiences is used for the update, performed as follows:

$$\nabla J(\theta) = \mathbb{E}_\pi \left[ \alpha \cdot \left( R^* \cdot \frac{\nabla \pi(a|s, \theta)}{\pi(a|s, \theta)} + \beta \cdot H_\pi(\cdot, s) \right) \right] \quad (5)$$

---

**Algorithm 1:** Policy learning in CONQUER

**Input:** question $q_t$, KG $K$, step size ($\alpha > 0$), number of rollouts ($rollouts$), size of update ($batchSize$), entropy weight ($\beta$)
**Output:** updated policy parameters $\theta$
▷ On initial call: Initialize $\theta$ randomly
1   $q_t^{cxt} \leftarrow loadContext(q_t)$
2   $E_t^{cxt} = detectContextEntities(q_t^{cxt}, q_t)$
3   **foreach** $e_t^{cxt} \in E_t^{cxt}$ **do**
4     $\mathcal{P} \leftarrow getKGPaths(e_t^{cxt}, K)$
5     $s \leftarrow (q_t^{cxt}, q_t, e_t^{cxt})$
6     $q_t \leftarrow BERT(q_t), q_t^{cxt} \leftarrow BERT(q_t^{cxt})$
7     $A_s \leftarrow stack(BERT(p))$, where $p \in \mathcal{P}$
8     $P(A_s) = \sigma(A_s \times (W_2 \times ReLU(W_1 \times [q_t^{cxt}; q_t])))$
9     $count \leftarrow 0$
10    **while** $count < rollouts$ **do**
11      $i \sim Categorical(P(A_s))$
12      $a \leftarrow A_s^i$, where $a := p_i$ and $(e_t^{cxt}, p_i, e^{ans}) \in K$
13      $q_{t+1} \leftarrow getUserFeedback(e^{ans})$
14      $q_{t+1}^{cxt} \leftarrow loadContext(q_{t+1})$
15      $s_{next} \leftarrow (q_{t+1}^{cxt}, q_{t+1}, e^{ans})$
16      **if** $isReformulation(q_t, q_{t+1})$ **then** $R \leftarrow -1$
17      **else** $R \leftarrow 1$
18      $experience.enqueue(s, a, s_{next}, R)$
19      $count \leftarrow count + 1$
20    **end**
21 **end**
22 **if** $|experience| >= batchSize$ **then**
23    $updateList \leftarrow experience.dequeue(batchSize)$
24    $batchUpdate \leftarrow 0, RList \leftarrow getRewards(updateList)$
25    $\bar{R} \leftarrow mean(RList), \sigma_R \leftarrow std\_dev(RList)$
26    **foreach** $(s, a, s_{next}, R) \in updateList$ **do**
27      $H_\pi(\cdot, s) \leftarrow -\sum_{a \in A_s} \pi(a|s) \cdot log\pi(a|s)$
28      $R^* \leftarrow \frac{R - \bar{R}}{\sigma_R}$
29      $batchUpdate \leftarrow batchUpdate + R^* \frac{\nabla \pi(a|s, \theta)}{\pi(a|s, \theta)} + \beta H_\pi(\cdot, s)$
30    **end**
31    $\theta \leftarrow \theta + \alpha \cdot batchUpdate$
32 **return** $\theta$

---

where $\alpha$ is the step size, $R^*$ is the normalized return, $\beta$ is a weighting constant for $H_\pi(\cdot, s)$ [8, 15], that is an entropy regularization term:

$$H_\pi(\cdot, s) = -\sum_{a \in A_s} \pi(a|s) \cdot log \, \pi(a|s) \quad (6)$$

which is added to the update to ensure better exploration and prevent the agent from getting stuck in local optima.

Parameters $\theta$ are updated in the direction that increases the probability of taking action $a$ again when seeing $s$ next time. The update is inversely proportional to the action probability to not favor frequent actions (see [59], Chapter 13, for more details). Finally, we normalize each reward by subtracting the mean and by dividing by the standard deviation of all rewards in the current batch update: $R^* = \frac{R - \bar{R}}{\sigma_R}$ to reduce variance in the update. **Algorithm 1** shows high-level pseudo-code for the policy learning in CONQUER. We now describe how we obtain the rewards used in the update.

## 4.3 Predicting reformulations

Each answer entity $e^{ans}$ reached by an agent after taking the sampled action is presented to the user. An ideal user, according to our assumption, would ask a follow-up question that is either a reformulation of the same intent (if she thinks the answer is wrong), or an expression of a new intent (if the answer seems correct). This sequence of the original question and the follow-up is then passed on to a reformulation detector to decide whether the two questions express the same intent. We devise such a predictor by fine-tuning a BERT model for sentence pair classification [17] on a large set of such question pairs. Based on this prediction (reformulation or not), we deduce if the generated answer entity $e^{ans}$ was correct (no reformulation) or not (reformulation). The agent then receives a positive reward (+1) when the answer was correct and a negative one (-1) otherwise.

## 5 GENERATING ANSWERS

The learned policy can now be used to generate answers for conversational questions. This happens in two steps: i) selecting actions by individual agents to reach candidate answers and ii) ranking the candidates to produce the final answer.

**Selecting actions**. Given a question $q_i$ at turn $t$, the first step is to extract the context entities $\{e_t^{cxt}\}$. Usually there are several of these and, therefore, several starting points for RL walks. Multiple agents traverse the KG in parallel, based on the predictions coming from the trained policy network. Each agent takes the top-$k$ predicted actions ($k$ is typically small, five in our experiments) from the policy network greedily (no explorations at answering time).

**Ranking answers**. Agents land at end points after following actions predicted for them by the policy network. These are all candidate answers $\{e^{ans}\}$ that need to be ranked. We interpret the probability scores coming from the network (associated with the predicted action) as a measure of the system's confidence in the answer and use it as our main ranking criterion. When multiple agents land at the same end point, we use this to boost scores of the respective candidates by adding scores for the individual actions. Candidate answers are then ranked by this final score, and the top-1 entity is shown to the user.

## 6 BENCHMARK WITH REFORMULATIONS

None of the popular QA benchmarks, like WebQuestions [5], ComplexWebQuestions [60], QALD [63], LC-QuAD [19, 61], or CSQA [52], contain sessions with questions with reformulations by real users. The only publicly available benchmark for ConvQA over KGs based on a real user study is ConvQuestions [12]. Therefore, we used conversation sessions in ConvQuestions as input to our own user study to create our benchmark ConvRef.

**Workflow for user study**. Study participants interacted with a baseline ConvQA system. In this way, we were able to collect real reformulations issued in response to seeing a wrong answer, rather than static paraphrases. To create such a baseline system we trained our randomly initialized policy network with simulated reformulation chains. ConvQuestions comes with one interrogative paraphrase per question, which could be viewed as a weak proxy for reformulations. A paraphrase was triggered as reformulation in case of a wrong answer during training.

| Nature of reformulation | Percentage |
|---|---|
| Words were replaced by synonyms | 15% |
| Expected answer types were added | 14% |
| Coreferences were replaced by topic entity | 24% |
| Whole question was rephrased | 71% |
| Words were reordered | 5% |
| Completed a partially implicit question | 20% |

**Table 2: Types of reformulations in ConvRef. Each reformulation may belong to multiple categories.**

The conversations shown to the users were topically grounded in the 350 seed conversations in ConvQuestions. Since we have paraphrases for each question, we also use the conversations where the original questions are replaced by their paraphrased version. This way we obtain 700 conversations, each with 5 turns. Users were shown follow-up questions in a given conversation interactively, one after the other, along with the answer coming from the baseline QA system. For wrong answers, the user was prompted to reformulate the question up to four times if needed. In this way, users were able to pose reformulations based on previous wrong answers and the conversation history. Note that the baseline QA system often gave wrong answers, as our RL model had not yet undergone full training. This provided us with a challenging stress-test benchmark, where users had to issue many reformulations.

**Participants**. We hired 30 students (Computer Science graduates) for the study. Each participant annotated about 23 conversations. The total effort required 7 hours per user (including a 1-hour briefing), and each user was paid 10 Euros per hour (comparable to AMT Master workers). The final session data was sanitized to comply with privacy regulations.

**Final benchmark**. The final ConvRef benchmark was compiled as follows. Each information need in the $11.2k$ conversations from ConvQuestions is augmented by the reformulations we collected. This resulted in a total of $262k$ turns with about $205k$ reformulations. We followed ConvQuestions ratios for the train-dev-test split, leading to $6.7k$ training conversations and $2.2k$ each for dev and test sets. While participants could freely reformulate questions, we noticed different patterns in a random sample of 100 instances (see Table 2). Examples of reformulations are shown in Table 3. Reformulations had an average length of about 7.6 words, compared to about 6.7 for the initial questions per session. The complete benchmark is available at https://conquer.mpi-inf.mpg.de.

## 7 EXPERIMENTAL FRAMEWORK

### 7.1 Setup

**KG**. We used the Wikidata NTriples dump from 26 April 2020 with about $12B$ triples. Triples containing URLs, external IDs, language tags, redundant labels and descriptions were removed, leaving us with about $2B$ triples. The data was processed according to Sec. 2.1 and loaded into a Neo4j graph database (https://neo4j.com). All data resided in main memory (consuming about 35 GB, including indexes) and was accessed with the Cypher query language.

**Context entities**. We used ELQ [34] to obtain NED scores. Frequency clip $f_{max}$ was set to 100. Parameters $h_1, h_2, h_3, h_4, h_{cxt}$ were tuned on our development set of 100 manually annotated utterances and set to 0.1, 0.1, 0.7, 0.1, 0.25, respectively (see Sec. 3).

| |
|---|
| **Original question:** *in what location is the movie set?* **[Movies]** |
| **Wrong answer:** `Doctor Sleep` |
| **Reformulation:** *where does the story of the movie take place?* |
| **Original question:** *which actor played hawkeye?* **[TV Series]** |
| **Wrong answer:** `M*A*S*H Mania` |
| **Reformulation:** *name of the actor who starred as hawkeye?* |
| **Original question:** *release date album?* **[Music]** |
| **Wrong answer:** `01 January 2012` |
| **Reformulation:** *on which day was the album released?* |
| **Original question:** *what's the first one?* **[Books]** |
| **Wrong answer:** `Agatha Christie` |
| **Reformulation:** *what is the first miss marple book?* |
| **Original question:** *who won in 2014?* **[Soccer]** |
| **Wrong answer:** `NULL` |
| **Reformulation:** *which country won in 2014?* |

Table 3: Sample reformulations from CONVREF.

**RL and neural modules**. The code for the RL modules was developed using the TensorFlow Agents library (https://www.tensorflow.org/agents). When the number of KG paths for a context entity exceeded 1000, a thousand paths were randomly sampled owing to memory constraints. All models were trained for 10 epochs, using a batch size of 1000 and 20 rollouts per training sample. All reported experimental figures are averaged over five runs, resulting from differently seeded random initializations of model parameters. We used the Adam optimizer [31] with an initial learning rate of 0.001. The weight $\beta$ for the entropy regularization term is set to 0.1. We used an uncased BERT-base model (https://huggingface.co/bert-base-uncased) for obtaining encodings of $\{q, q^{cxt}, a\}$. To obtain encodings of a sequence, two averages were performed: once over all hidden layers, and then over all input tokens. Dimension $d = 768$ (from BERT models), and accordingly sizes of the weight matrices were set to $W_1 = |input| \times 768$ and $W_2 = 768 \times 768$, where $|input| = d$ or $|input| = 2d$ (in case we prepend context questions $q_t^{cxt}$).

**Reformulation predictor**. To avoid any possibility of leakage from the training to the test data, this classifier was trained only on the CONVREF dev set (as a proxy for an orthogonal source). We fine-tuned the sequence classifier at https://bit.ly/2OcpNYw for our sentence classification task. Positive samples were generated by pairing questions within the same intent, while negative sampling was done by pairing across different intents from the *same conversation*. This ensured lexical overlap in negative samples, necessary for more discriminative learning.

## 7.2 CONQUER configurations

The CONQUER method has four different setups for training, that are evaluated at answering time. These explore two orthogonal sources of noise and stem from two settings for the user model and two for the reformulation predictor.

**User model:**
Although CONVREF is based on a user study, we do not have continuous access to users during training. Nevertheless, like many other Monte Carlo RL methods, we would like to simulate user behavior for accumulating more interactions that could enrich our

training (for example, by performing rollouts). We thus define ideal and noisy user models as follows.

- **Ideal:** In an ideal user model, we assume that users always behave as expected: reformulating when the generated answer is wrong and only moving on to a new intent when it is correct. Since each intent in the benchmark is allowed up to 5 times, we loop through the sequence of user utterances within the same intent if we run out of reformulations.
- **Noisy:** In the noisy variant, the user is free to move on to a new intent even when the response to the last turn was wrong. This models realistic situations when a user may simply decide to give up on an information need (out of frustration, say). There are at most 4 reformulations per intent in CONVREF: so a new information need may be issued after the last one, regardless of having seen a correct response.

**Reformulation predictor:**
- **Ideal:** In an ideal reformulation predictor, we assume that it is known upfront whether a follow-up question is a reformulation or not (from annotations in CONVREF).
- **Noisy:** In the noisy predictor, we use the BERT-based reformulation detector which may include erroneous predictions.

## 7.3 Baseline

We use the CONVEX system [12] as the state-of-the-art ConvQA baseline in our experiments. CONVEX detects answers to conversational utterances over KGs in a two-stage process based on judicious graph expansion: it first detects so-called frontier nodes that define the context at a given turn. Then, it finds high-scoring candidate answers in the vicinity of the frontier nodes. Hyperparameters of CONVEX were tuned on the CONVREF train set for fair comparison.

## 7.4 Metrics

Answers form a ranked list, where the number of correct answers is usually one, but sometimes two or three. We use three standard metrics for evaluating QA performance: i) precision at the first rank (P@1), ii) answer presence in the top-5 results (Hit@5) and iii) mean reciprocal rank (MRR). We use the standard metrics of i) precision, ii) recall and iii) F1-score for evaluating context entity detection quality. These measures are also used for assessing reformulation prediction performance, where the output is one of two classes: reformulation or not. Gold labels are available from CONVREF.

## 8 RESULTS AND INSIGHTS

## 8.1 Key findings

Tables 4 and 5 show our main results on the CONVREF test set. P@1, Hit@5 and MRR are measured over distinct intents, not utterances. For example, even when an intent is satisfied only at the third reformulation, we deem P@1 = 1 (and 0 when the correct answer is not found after five turns). The effort to arrive at the answer is measured by the number of reformulations per intent (*"RefTriggers"*) and by the number of intents satisfied within a given number of reformulations (*"Ref = 1"*, *"Ref = 2"*, ...). Statistical significance tests are performed wherever applicable: we used the McNemar's test for binary metrics (P@1, Hit@5) and the $t$-test for real-valued ones (MRR, F1). Tests were unpaired when there are unequal numbers

| Method | P@1 | Hit@5 | MRR | RefTriggers | Ref = 0 | Ref = 1 | Ref = 2 | Ref = 3 | Ref = 4 |
|---|---|---|---|---|---|---|---|---|---|
| Conquer IdealUser-IdealReformulationPredictor | 0.339 | 0.426 | 0.376 | 30058 | **3225** | 292 | 154 | 70 | 56 |
| Conquer IdealUser-NoisyReformulationPredictor | 0.338 | **0.429** | 0.377 | 30358 | 3099 | 338 | 170 | 79 | 100 |
| Conquer NoisyUser-IdealReformulationPredictor | **0.353** | 0.428 | **0.387** | 29889 | 3163 | 403 | 187 | **90** | **116** |
| Conquer NoisyUser-NoisyReformulationPredictor | 0.335 | 0.417 | 0.370 | 30726 | 2913 | **425** | **216** | **90** | 104 |
| Convex [12] | 0.225 | 0.257 | 0.241 | 34861 | 1980 | 278 | 200 | 24 | 35 |

Table 4: Main results on answering performance over the ConvRef test set. All Conquer variants outperformed Convex with statistical significance, and required less reformulations than Convex to provide the correct answer.

| Conquer/Baseline | Movies | TV Series | Music | Books | Soccer |
|---|---|---|---|---|---|
| IdealUser-IdealRef | 0.320 | 0.316 | 0.281 | **0.449** | **0.329** |
| IdealUser-NoisyRef | 0.344 | 0.340 | 0.303 | 0.425 | 0.308 |
| NoisyUser-IdealRef | **0.368** | **0.367** | **0.324** | 0.413 | **0.329** |
| NoisyUser-NoisyRef | 0.327 | 0.296 | 0.300 | 0.381 | 0.327 |
| Convex [12] | 0.274 | 0.188 | 0.195 | 0.224 | 0.244 |

Table 5: P@1 results across topical domains.

| All | No Overlap | No Match | No NED | No prior |
|---|---|---|---|---|
| **0.731** | 0.726 | 0.728 | 0.684 | 0.718 |

Table 6: Ablation study for context entity detection (F1).

| Context model | P@1 | Hit@5 | MRR |
|---|---|---|---|
| Curr. ques. + Cxt. ent. | **0.294** | **0.407** | **0.346** |
| Curr. ques. + Cxt. ent. + First ques. | 0.254 | 0.370 | 0.305 |
| Curr. ques. + Cxt. ent. + First ques. + Prev. ques. | 0.257 | 0.370 | 0.307 |
| Curr. ques. + Cxt. ent. + First refs. + Prev. refs. | 0.262 | 0.382 | 0.316 |

Table 7: Effect of context models on answering performance.

of utterances handled in each case due to unequal numbers of reformulations triggered and paired in other standard cases. 1-sided tests were performed for checking for superiority (for baselines) or inferiority (for ablation analyses). In all cases, null hypotheses were rejected when $p \leq 0.05$. Best values in table columns are marked in **bold**, wherever applicable.

**Conquer is robust to noisy models**. We did not observe any major differences among the four configurations of Conquer. Interestingly, some of the noisy versions (*"IdealUser-NoisyRef"* and *"NoisyUser-IdealRef"*) even achieve the absolute best numbers on the metrics, indicating that a certain amount of noise and non-deterministic user behavior may in fact help the agent to generalize better to unseen conversations. Note that the variants with ideal models are not to be interpreted as potential upper bounds for QA performance: while *"IdealUser"* represents a model of systematic user behavior, *"IdealRef"* rules out one source of model error.

**Conquer outperforms Convex**. All variants of Conquer were significantly better than the baseline Convex on the three metrics ($p < 0.05$, 1-sided tests), showing that Conquer successfully learns from a sequence of reformulations. Convex on the other hand, as well as any of the existing (conversational) KG-QA systems [23, 52, 55], cannot learn from incorrect answers and indirect signals such as reformulations. Additionally, Conquer can also be applied in the standard setting were <question, correct answer> instances are available. When trained on the original ConvQuestions benchmark, that contains gold answers but lacks reformulations, Conquer achieves P@1=0.263, Hit@5=0.343 and MRR=0.298, again outperforming Convex with P@1=0.184, Hit@5=0.219, MRR=0.200.

**Conquer needs fewer reformulations**. In Table 4, *"RefTriggers"* shows the number of reformulations needed to arrive at a correct answer (or reaching the maximum of 5 turns). We observe that Conquer triggers substantially fewer reformulations ($\simeq 30k$) than Convex ($\simeq 34k$). This confirms an intuitive hypothesis that when a model learns to answer better, it also satisfies an intent faster (less turns needed per intent). Zooming into this statistic (*"Ref = 0, 1, 2, ..."*), we observe that Conquer answers a bulk of the intents without needing any reformulation, a testimony to successful training ($\simeq 3k$ in comparison to $\simeq 2k$ for Convex). The numbers quickly taper off with subsequent turns in a conversation, but remain higher than the baseline. Convex relies on a context graph that is iteratively

expanded over turns; this often becomes unwieldy at deeper turns. Unlike Convex, we found Conquer's performance to be relatively stable even for higher intent depths (P@1 = 0.237, 0.397, 0.336, 0.319, 0.385 for intents 1 through 5, respectively).

**Conquer works well across domains**. P@1 results for the five topical domains in ConvQuestions are shown in Table 5. We note that the good performance of Conquer is not just by skewed success on one or two favorable domains (while being significantly better than Convex for each topic), but holds for all five of them (books being slightly better than the others).

## 8.2 In-depth analysis

Having shown the across-the-board superiority of Conquer over Convex, we now scrutinize the various components and design choices that make up the proposed architecture. All analysis experiments are reported on the ConvRef dev set, and the realistic *"NoisyUser-NoisyRef"* Conquer variant is used by default.

**All features vital for context entity detection**. We first perform an ablation experiment (Table 6) on the features responsible for identifying context entities. Observing F1-scores averaged over questions, it is clear that all four factors contribute to accurately identifying context entities (no NED as well as no prior scores resulted in statistically significant drops). It is interesting to understand the trade-off here: a high precision indicates a handful of accurate entities that may not create sufficient scope for the agents to learn meaningful paths. On the other hand, a high recall could admit a lot of context entities from where the correct answer may be reachable but via spurious paths. The F1-score is thus a reliable indicator for the quality of a particular tuple of hyperparameters.

**Context entities effective in history modeling**. After examining features for $E^{cxt}$, let us take a quick look at the effect of $Q^{cxt}$. We tried several variants and show the best three in Table 7. While the Conquer architecture is meant to have scope for incorporating various parts of a conversation, we found that explicitly encoding previous questions significantly degraded answering quality (the first row, where $Q^{cxt} = \phi$, works significantly better than all other

options). *"refs."* indicate that embeddings of reformulations for that intent were averaged. Without *"refs."*, only the first question in that intent is used. Results indicate that context entities from the KG suffice to create a satisfactory representation of the conversation history. Note that these $E_t^{cxt}$ are derived not just from the current turn, but are carried over from previous ones. Nevertheless, we believe that there is scope for using $Q^{cxt}$ better: history modeling for ConvQA is an open research topic [24, 45, 48, 49] and reformulations introduce new challenges here.

**Reformulation predictor works well**. A crucial component of the success of Conquer's *"NoisyRef"* variants is the reformulation detector. Due to its importance, we explored several options like fine-tuned BERT [17] and fine-tuned RoBERTa [36] models to perform this classification. RoBERTa produced slightly poorer performance than BERT, which was really effective (Table 8). Prediction of new intents is observed to be slightly easier (higher numbers) due to expected lower levels of lexical overlaps.

**Path label preferable as actions**. When defining what constitutes an action for an agent, we have the option of appending the answer entity $e^{ans}$ or the context entity $e^{cxt}$ to the KG path (world knowledge in BERT-like encodings often helps in directly finding good answers). We found, unlike similar applications in KG reasoning [16, 35, 46], excluding $e^{ans}$ actually worked significantly better for us (Table 9, row 1 vs. 2). This can be attributed to the low semantic similarity of answer nodes with the question, that acts as a confounding factor. Including $e^{cxt}$ does not change the performance (row 1 vs. 3). The reason is that an agent selects actions starting at one specific start point ($e^{cxt}$): all of these paths thus share the embedding for this start point, resulting in an indistinguishable performance. The last row corresponds to matching the question to the entire KG fact, which again did not work so well due to the same distracting effect of the answer entity $e^{ans}$.

**Error analysis points to future work**. We analyzed 100 random samples where Conquer produced a wrong answer (P@1 = 0). We found them to be comprised of: 17% ranking errors (correct answer in top-5 but not at top-1), 23% action selection errors (context entity correct but path wrong), 30% context entity detection errors (including 3% empty set), 23% not in the KG (derived quantitative answers), and 7% wrong gold labels.

**Answer ranking robust to minor variations**. Our answer ranking (Sec. 5) uses cumulative prediction scores (scores from multiple agents added), with P@1 = 0.294. We explored variants where we used prediction scores with ties broken by majority voting since an answer is more likely if more agents land on it (P@1 = 0.291), majority voting with ties broken with higher prediction scores (P@1 = 0.273), and taking the candidate with the highest prediction score without majority voting (P@1 = 0.294). Most variants were nearly equivalent to each other, showing *robustness of the learnt policy*.

**Runtimes**. The policy network of Conquer takes about 10.88ms to produce an answer, as averaged over test questions in ConvRef. The maximal answering time was 1.14s.

## 9 RELATED WORK

**QA over KGs**. KG-QA has a long history [5, 53, 62, 69], evolving from addressing simple questions via templates [1, 4] and neural methods [27, 70], to more challenging settings of complex [7, 37], heterogeneous [42, 58] and conversational QA [39, 55]. Recent work

| Method | Fine-tuned BERT | | | Fine-tuned RoBERTa | | |
|---|---|---|---|---|---|---|
| Class | Prec | Rec | F1 | Prec | Rec | F1 |
| New intent | 0.986 | 0.944 | 0.965 | 0.988 | 0.924 | 0.955 |
| Reformulations | 0.810 | 0.948 | 0.873 | 0.760 | 0.956 | 0.847 |

**Table 8: Classification performance for reformulations.**

| Method | P@1 | Hit@5 | MRR |
|---|---|---|---|
| Path | **0.294** | 0.407 | **0.346** |
| Path + Answer entity | 0.275 | 0.394 | 0.329 |
| Context entity + Path | 0.293 | **0.408** | **0.346** |
| Context entity + Path + Answer entity | 0.273 | 0.398 | 0.328 |

**Table 9: Design choices of actions for the policy network.**

on ConvQA in particular includes [12, 23, 52, 55]. However, these methods do not consider question reformulations in conversational sessions and solely learn from question-answer training pairs.

**RL in KG reasoning**. RL has been pursued for KG reasoning [15, 22, 35, 57, 67]. Given a relational phrase and two entities, one has to find the best KG path that connects these entities. This paradigm has been extended to multi-hop QA [46, 73]. While Conquer is inspired by some of these settings, the ConvQA problem is very different, with multiple entities from where agents could potentially walk and missing entities and relations in the conversational utterances.

**Reformulations**. In the parallel field on text-QA, question reformulation or rewriting has been pursued as conversational question completion [3, 51, 64, 68, 71]. In our work, we revive the more traditional sense of reformulations [10, 13, 25, 28], where users pose queries in a different way when system responses are unsatisfactory. Several works on search and QA apply RL to automatically generate or retrieve reformulations that would proactively result in the best system response [8, 14, 41, 44]. In contrast, Conquer learns from free-form user-generated reformulations. Question paraphrases can be considered as proxies of reformulations, without considering system responses. Paraphrases have been leveraged in a number of ways in QA [6, 18, 20]. However, such models ignore information about sequences of user-system interactions in real conversations.

**Feedback in QA**. Incorporating user feedback in QA is still in its early years [2, 9, 32, 72]. Existing methods leverage positive feedback in the form of user annotations to augment the training data. Such explicit feedback is hard to obtain at scale, as it incurs a substantial burden on the user. In contrast, Conquer is based on the more realistic setting of implicit feedback from reformulations, which do not intrude at all on the user's natural behavior.

## 10 CONCLUSION

This work presented Conquer: an RL-based method for conversational QA over KGs, where users pose ad-hoc follow-up questions in highly colloquial and incomplete form. For this ConvQA setting, Conquer is the first method that leverages implicit negative feedback when users reformulate previously failed questions. Experiments with a benchmark based on a user study showed that Conquer outperforms the state-of-the-art ConvQA baseline Convex [12], and that Conquer is robust to various kinds of noise.

**Acknowledgments**. We would like to thank Philipp Christmann from MPI-Inf for helping us with the experiments with CONVEX.

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
