# OpenReview forum: "Reinforcement Learning from Reformulations in Conversational Question Answering over Knowledge Graphs"
_ACM.org/SIGIR/Badging_

### Official Review · ~Antonio_Mallia2 · 2021-09-09
**Artifact evaluated as Functional**

**Comment:**

Dear Magdalena Kaiser, Rishiraj Saha Roy, Gerhard Weikum, Thank you for your submission!

Overall, I do not see any issues awarding the "Functional" badge.
I've tested everything on an Ubuntu 18.04.4 LTS without any GPUs installed.

Functional
"The artifacts associated with the research are found to be"

"documented"
(+) The paper has a web page dedicated (https://conquer.mpi-inf.mpg.de/) where the paper is described and there is a link to the source code repository
(+) The repository contains a detailed README file.
(+) The code is  properly documented

"consistent"
(+) The pruned data complies with the descriptions provided in the paper.
(+) The code repository includes the corresponding scripts for training/evaluation/etc.

"complete"

(+) All scripts for training, conducting and evaluating the experiments are included. The benchmark, all required intermediate data and main results are publicly available.

"exercisable"
(+) The scripts run without errors and produce results.
(+) The required packages and versions are documented in requirements.txt

"the artifact should be made available in an online repository; the artifact corresponds to what is mentioned in the corresponding paper"
(+) The code is hosted on GitHub and is released with MIT License. The data is hosted on https://conquer.mpi-inf.mpg.de/static/data.zip and is retrievable at the time of review.



**Awarded Badges:**

["Artifacts Evaluated – Functional"]

---

> ### Public Comment · ~Magdalena_Kaiser1 · 2021-09-15
> **Feedback regarding "Reusable and Available" badge?**
>
> Dear Antonio Mallia,
>
> thanks a lot for your review regarding the "Functional" badge. Do you also have feedback for us regarding the "Reusable and Available" badge?
>
> Kind Regards,
> Magdalena

---

### Official Review · ~Diego_Ceccarelli1 · 2021-10-03
**Artifact Resuable and Available**

**Comment:**

I confirm that we trained the model and reproduced the results indicated in the paper (for one of the experiments).
I've tested everything on an Ubuntu 18.04.4 LTS without any GPUs installed.

For the badge: "Artifact Resuable and Available":

In my opinion these steps are satisfied:

- Datasets have description of their “schema”
- There is an explanation of the intended way to process them;
- The code itself need to be properly commented and documented, e.g. Javadoc or similar for other languages;
- All the materials used to conduct the study should be available, e.g. instructions, questionnaires, interview questions, scales, qualitative analysis code-books, or study protocols.
- The content of the artifact corresponds to its description in the paper, that it may produce the results reported in the paper, and that no part of it is missing, relative to the results reported in the paper.
- code  can be successfully compiled and executed. Source code must always be available but, on top of this, it is strongly suggested that the artifact is also packaged as a virtual machine or live notebook to enable the easiest possible fruition of the artifact (see section on packaging below).
- The guide then walks the reviewer through the deployment process, specifying every command to be executed until the system is fully deployed. In particular, the guide must not presume any prior knowledge, and the steps must be effortless on the part of the reviewer (i.e., not some instruction like “Install and configure Redis”, but the actual commands to do it the way the authors need it).
- The expected review effort for this type of badge is in the order of very few hours and more time is considered an indicator of potential issues with the artifact.

These steps need to be addressed:

1.
- The README of the code repository contains a step-by-step deployment guide. The starting point of the guide is a freshly installed OS (say, a numbered version of Ubuntu LTS version).

The OS is not specified. we used Ubuntu 18.04.4 LTS without any GPUs installed.
Please note that it was not possible to run this on a OSX because a dependency was not available for the os in Python. The authors made a good job documenting how to install all the components but it will still require time to reuse everything.  I would suggest the authors to have a look at Docker and consider to provide a Docker image with everything installed in the future. For this badge I think it's fine to just specify that the code runs under Linux.

2.
Authors provide a dataset but I haven't seen in the readme documentation/code on how to load it, as specified in the guideline:

- a default parser is provided with the artifact.
- datasets can be successfully unpacked and parsed.
- For datasets to be reusable, the reviewer should really ensure they can import them (or a subset by some criteria) into some data analysis totool either via pre-processing (or directly) and then carry out some basic analysis – producing graphs, histograms etc.
In particular, the paper:
	▪	Gebru, T., Morgenstern, J., Vecchione, B., Wortman Vaughan, J., Wallach, H., Daumeé III, H., and Crawford, K. (2018). Datasheets for Datasets. arXiv.org, Databases (cs.DB), arXiv:1803.09010 (https://arxiv.org/abs/1803.09010).
contains best practices that should be adopted by authors in packaging and describing their datasets.

There should be a single class that allows to load the dataset (maybe in a pandas dataframe?) so that other researchers can easily use it from the code and run analysis. How to load the dataset should be documented into the README.

Once 1. and 2. are addressed I would be happy to approve the badge.

Best regards,
Diego



**Awarded Badges:**

["Artifacts Evaluated – Functional"]

---

> ### Public Comment · ~Magdalena_Kaiser1 · 2021-10-05
> **Steps Addressed**
>
> Dear Diego,
>
> thanks a lot for your review. I have updated the README:
>
> 1. Inserted a note that the code runs on Linux only. In the future, we will consider using Docker as you suggested.
> 2. Inserted comments on how to load/process the dataset.
>
> Best Regards,
> Magdalena

---

> > ### Public Comment · ~Diego_Ceccarelli1 · 2021-12-31
> > **Artifact Resuable and Available**
> >
> > Thanks Magdalena, I reviewed point 1. and 2. and I confirm that the artifact is reusable and available.

---

### Public Comment · Program_Chairs · 2021-12-08
**Functional, Reusable, and Available Badges**

According to the discussion you had with the reviewers and to the final reviews, your artifact is ready for badging.

We are happy to award you the following badges:
* Artifacts Evaluated – Functional
* Artifacts Evaluated – Reusable and Available

To have your artifact included in the ACM DL we need to received a single zip file containing it.

Could you, please, send it to us at the following email address:

aec_sigir@acm.org

at your earliest convenience?